# P-Wave Axis of Schoolchildren Who Were Once Breastfed

**DOI:** 10.3390/children10071255

**Published:** 2023-07-21

**Authors:** Juan-Antonio Costa, Carla Rodriguez-Trabal, Ignacio Pareja, Alicia Tur, Marianna Mambié, Mercedes Fernandez-Hidalgo, Sergio Verd

**Affiliations:** 1Department of Paediatrics, Ca’n Misses District Hospital, Corona Street, 07800 Eivissa, Spain; curune@yahoo.es; 2Department of Pediatrics, Son Espases University Hospital, Valldemossa Road, 07120 Palma de Mallorca, Spain; carlatrabal1997@gmail.com; 3La Vileta Surgery, Paediatric Unit, Department of Primary Care, Matamusinos Street, 07013 Palma de Mallorca, Spain; ji.pareja@prepersa.es (I.P.); aliciaesther.tur@ibsalut.es (A.T.); marianna.mambie@gmail.com (M.M.); mercedes.fernandezhidalgo@ibsalut.es (M.F.-H.); 4Balearic Institute of Medical Research (IdISBa), Valldemossa Road, 07120 Palma de Mallorca, Spain

**Keywords:** electrocardiography, infant formula, breastfeeding, body mass index

## Abstract

Background. It has been known for decades that breastfeeding leads to a lower risk of asthma, respiratory infections, or metabolic syndrome at school age. In addition, evidence is now accumulating on the influence of breast milk on the shape, volume, or function of the heart and lungs. Within this field of research into the effects of breast milk on the structure of the heart and lungs, we have set out to analyze the differential electrocardiographic characteristics of schoolchildren who were once breastfed. Method. This was an observational cross-sectional study, including 138 children aged 6 or 12 consecutively presenting to a well-child clinic between May and December 2022. Inclusion criteria. The ability to perform reproducible ECG records, the feasibility of weighing and measuring patient, and breastfeeding data collected from birth were used as the inclusion criteria. Results. Using the 40° cut-off value for the mean P-wave axis among schoolchildren, 76% of never-breastfed children in our sample have a P-wave axis in a more vertical position than the mean as compared to 58% of ever-breastfed children (OR: 2.25; 95% CI: 3.13–1.36); there was no other significant difference between infant feeding groups in somatometric characteristics or ECG parameters. Conclusion. We found a significant difference of the mean values of the P-wave axis between never- and ever-breastfed children. Although this report should be approached cautiously, these findings add to the renewed interest in discerning developmental interventions to improve cardiovascular health.

## 1. Introduction

It is a recognized fact that breastfeeding contributes to children’s cardiovascular and respiratory health in the short, medium, and long term. Regarding the respiratory tract, a 2018 meta-analysis reported that donor milk protects against bronchopulmonary dysplasia [1]; former and current research has found that infants who were breastfed had subsequently fewer lower tract respiratory hospitalizations than formula-fed infants [2]; a systematic review that summarizes the research on the link between breastfeeding and bronchiolitis from 2000 to 2021 shows that both exclusive and partial breastfeeding lower the severity of bronchiolitis [3]; and a 2021 meta-analysis restricted to cohort or randomized studies found that both any breastfeeding and exclusive breastfeeding are linked to a lower risk of getting childhood asthma [4]. In addition, emerging research suggests that more prolonged breastfeeding benefits lung function in later childhood, with the most consistent effect on increased lung growth and vital capacity [5].

On the other hand, the amount of evidence gathered during decades in the field of cardiometabolic risk protection through breastfeeding has established that a higher percentage or longer duration of human milk feeding is linked to lower blood pressure or healthier lipid profile in childhood [6], decreased cholesterol levels in adulthood, or lower carotid intima-media thickness with advancing age [7]. More recently, the expanding understanding of the role of human milk in the proper development of cardiac function and structure has provided insight into the mechanism by which early human milk exposure may have preventive effects by halting pathophysiological alterations potentially leading to cardiovascular disease [8]. In the same line of research, the neonatal period represents a critical window for cardiac maturational changes [9], during which breastfeeding has been shown to predict greater right and left ventricle volume index and stroke volume with lower pulmonary pressures [10].

We, therefore, hypothesized that pediatric ECG parameters might be affected by different human milk exposure in the early neonatal period.

## 2. Materials and Methods

### 2.1. Study Design and Population

The study was carried out at the Vileta pediatric unit, which belongs to an urban primary care surgery facility that attends roughly 26,000 people in Majorca, Spain. The original study was not designed specifically for analyzing the association between previous breastfeeding and current electrocardiogram (ECG) parameters of schoolchildren. The main ongoing study aims to identify children in the Balearic Islands at risk of malignant arrhythmia at the beginning and end of primary school. 

In particular, we carried-out an observational cross-sectional study including 138 children aged 6 or 12. All children these ages presenting to the to the Vileta well-child clinic between May and December 2022 were studied. The local institutional review board approved the protocol for the study. Inclusion criteria included age 6 or 12 years; ability to perform reproducible ECG records; and breastfeeding data collected at the first consultation with the pediatrician after discharge from the maternity hospital. Exclusion criteria included children with congenital or acquired heart disease. 147 children were invited to participate. Of these, six declined, and there were three cases of lack of feasibility or reproducibility of ECG recording. Data on somatometric variables were collected from previous and current medical records.

### 2.2. Ethics Requirements

The Regional Committee of the Balearic Islands for Medical and Health Research Ethics approved this study (ECIAP-2023).

### 2.3. ECG Methodology 

After obtaining informed consent, a standard ECG was performed in the lying supine position. At this visit, children had no recent history of respiratory illness and were in stable clinical condition. Following a 15 min period of rest in the supine position, participants underwent two 10 s standard 12-lead ECG recordings; one of two qualified pediatric nurses took the resting ECGs standardized at 25 mm/s and 10 mm/mV, using the PageWriter TC20 Cardiograph (Philips, Inc., Amsterdam, The Netherlands). ECGs that were of poor quality, were affected by severe motion artifact, or had inconsistencies with identification or lead placement were excluded. The participants were advised to remain silent throughout the recording while breathing normally. Each ECG parameter was automatically measured by the software.

### 2.4. Statistical Analysis

Differences in variables between never and ever breastfed children were examined using *t*-tests or Fisher’s exact tests. The data were checked for skewness by Kolmogorov–Smirnov tests of normality. Continuous data were presented as mean and standard deviation. Categorical data were presented as numbers and percentages. For every ECG parameter, descriptive statistics were determined for each feeding subgroup and for each somatometric variable. Bivariate analyses were conducted to find out the relationship between feeding type and each ECG parameters. Potential confounding factors included height, weight, or body mass index percentiles according to percentiles of Spanish growth standards. In all cases, significance was acknowledged if the probability of a type 1 error was less than 0.05. Data were analyzed using Microsoft Excel 365 Data Analysis Tool.

## 3. Results

### 3.1. Characteristics of Study Population

Demographic characteristics are presented in Table 1. In brief, they were 138 children of ages 6 or 12 years old (corresponding to two grades of primary school), weight from 14 to 70 kg, height from 95 to 175 cm, and body mass index from 10 to 28 kg/m^2^. In the study population, there were 74 girls and 65 boys. Overall, 75.5% of mothers had breastfed, and a descriptive evaluation of the prevalence of breastfeeding at the different time points is shown in Table 1. The median duration of any breastfeeding in 99 mother-child dyads, for which data were available, was 183 days (range: 1–2190). Breastfeeding can take many patterns; in our study, “any breastfeeding” includes several cases of extended breastfeeding because we accept breastfeeding or feeding babies mother’s milk away from their mother’s breasts plus supplements of infant formula or liquid or solid food.

### 3.2. Association between Any Breastfeeding and ECG or Somatometric Characteristics

In the study population, there was no significant difference between ever- and never-breastfed children in age (8.8 ± 3.0 vs. 9.3 ± 3 years, *p* = 0.50) or heart rate (83 ± 14 vs. 87 ± 11 beats/min, *p* = 0.72). Similarly, there was no significant difference between infant feeding groups in somatometric characteristics or ECG parameters, except for the P-wave axis.

### 3.3. P-Wave Axis Description

There is a statistically significant difference in the mean values of the P-wave axis between never and ever-breastfed children (47.12° ± 17.09° and 37.99° ± 24.53°, respectively; *p* = 0.048). Furthermore, using the 40° cut-off value for the mean P-wave axis among schoolchildren, 76% of never-breastfed children in our sample have a P-wave axis more vertical than the mean as compared to 58% ever-breastfed children (OR: 2.25; 95% CI: 3.13–1.36). Finally, there was no significant correlation between the P-wave axis or other ECG parameters that approach statistical association to infant feeding type with the somatometric characteristics of our population (Table 2).

On the other hand, we found no correlation between days of any breastfeeding and degrees of the P axis, Pearson’s correlation coefficient R = 0.061, *p*-value = 0.552.

## 4. Discussion

We have found a verticalization of the P-wave axis in primary school children who were never breastfed. Recent research on over 22,000 nonselected healthy children recruited from primary schools has determined that the mean P-wave axis is 40.4 degrees [11]. We report that the mean P-wave axis of ever-breastfed children in our sample was 37.9 degrees, while the same parameter in children who were never breastfed was 47.1 degrees (*p* = 0.04). To our knowledge, this is the pioneer descriptive study to establish a significant link between breastfeeding and the P-wave axis of schoolchildren.

This relation could not be explained by anthropometric differences or body position at the time of electrocardiographic recordings. A recent review of the ECG characteristics of obese children [12] found shifts of P-wave, QRS, and T-wave axes in overweight and obese children. However, there were no significant differences in body mass index between the ever- and never-breastfed children of our sample. Since changes in body position may result in significant P-wave shifts [13], all P-wave indices of participants were derived from resting, and standard 12-lead ECGs were taken in the supine position. In addition, current technology allows for excellent repeatable measurement of the P-wave axis [14].

Although the biological mechanisms underlying the effects of breastfeeding on this ECG parameter are unclear, two fundamental mechanisms may compete to explain the link between infant feeding and P-wave standards. 

Firstly, the role of infant feeding in preventing cardiovascular consequences of the metabolic syndrome has been investigated previously in patients of all ages. Studies show that breast-feeders have lower blood pressure than non-breast-feeders and a favorable lipid profile in childhood and adolescence [7,15]. Similarly, there is sufficient evidence that breastfeeding has a beneficial effect on cardiovascular fitness; different systematic reviews have confirmed that breastfed children have better maximal oxygen consumption index (VO_2_ max) and immediate or explosive skeletal muscle response than those who were never breastfed [16,17]. On the other hand, the data regarding the medium-term effects of human milk feeding on heart shape or cardiovascular function is at its early stage, but, taken together, these studies provides further evidence supporting the cardioprotective effects of early life environmental factors, including early breast-milk exposure. Breastfeeding improves microvascular function in children aged 11–14 [18]. In addition, children who were never breastfed tend to have narrower retinal venular calibers [19]. Newborn infants fed exclusively human milk will have increase ventricular stroke volume at one year [9]. Although the mechanism through which mother’s milk may modulate the cardiac phenotype in the long term is one of the missing pieces of this research, some authors stress that the reason why breastfeeding exerts these beneficial effects may lie in the growth factors contained in breast milk, which act precisely at critical times during which there are notable changes in the development of the heart and lungs [10]. 

Secondly, the effect of adults’ respiratory disease on the P-wave axis has been well studied since 1948 [20]. P-wave verticalization (>60°) has a strong inverse relationship with forced expiratory volume in 1 s (FEV1) and might also have a positive correlation with the radiological severity of lung hyperinflation [21]. In fact, *FEV1* is used to categorize the severity of asthma and adults’ chronic obstructive lung diseases. Regarding pediatric patients, most research has shown higher FEV1 in previously breastfed children compared to those who were never breastfed. Filippo et al. [22] found a dose-dependent association of breastfeeding duration with FEV1, Ogbuanu et al. [23] reported that FEV1 increased by 40 mL in children breastfed beyond four months, and Tennant et al. [24] showed that breastfeeding duration less than four weeks was a predictor of lower FEV1 among schoolchildren. Since the FEV1 is a flow governed by airway caliber and the elastic properties of lungs, it is inversely correlated with hyperinflation that is marked by reduced airway caliber [25]. The vertical P-wave axis reflects the atrial orientation in the thorax and is highly predictive of hyperinflation in children. A plausible explanation is that the right atrium, which is firmly attached to the diaphragm by a pericardial ligament, is moved inferiorly due to the diaphragm’s flattening, which causes a verticalization of the P-wave axis. An analysis of the pulmonary function of 102 children aged 6–18 years showed not only marked verticalization of the P-wave axis in children with moderate to severe hyperinflation but also that a vertical increase in the P-wave axis of >5° was highly sensitive of increased hyperinflation [20].

To summarize, our findings may result from complex interactions between the immunoactive and growth factors of human milk and the mechanical effect of suckling at the breast, or from a direct effect of breastfeeding on the growth of heart or lungs. In addition, recent research on its pathomorphological properties supports the oneness of the cardiorespiratory system.

Some limitations need to be addressed. First of all, this is a single-center study, the analysis is limited to an ECG recording at a single time point, and the small sample size reduces the generalizability of our results to the general population. Moreover, we found no correlation between days of any type of breastfeeding and degrees of the P-wave axis in schoolchildren when we know that a dose-response association would have strengthened a causal hypothesis. Second, details on breastfeeding type or duration are lacking. We analyze consequences of ever- versus never-breastfeeding. We acknowledge that we lack data on breastfeeding. However, a number of systematic reviews [26,27] on the different rate of childhood obesity or leukemia or other diseases between ever- and never-breastfed children is in our credit. Third, according to the bivariate analysis results, no potential confounders were considered. Fourth, residual confounding might be a distortion, as in any descriptive research. Last is the absence of an ECG evaluation at birth.

We acknowledge that randomization was not performed in our study and that this report should be approached with caution, but these findings add to the renewed interest in discerning developmental interventions aimed at improving cardiovascular health.

Strengths: Information on breastfeeding was recorded prospectively, which might exclude reporting recall bias. Given that the software automatically provided the P-wave axis as a single global measure, we assume the measurement error is constant across the whole range of values. In the last few years, research on pre-pathological findings is getting more significant in terms of Public Health. This includes slight, within normal ranges, but significant changes [12].

## 5. Conclusions

We report a significant clockwise rotation of above 15 grades in the mean P-wave axis of never-breastfed schoolchildren compared to their ever-breastfed peers.

## Figures and Tables

**Table 1 children-10-01255-t001:** Baseline characteristic and ECG parameters of included sample divided by infant feeding type.

Variables	Ever-Breastfed Schoolchildren N = 105	Never-Breastfed Schoolchildren N = 33	*p* Value
1. Any breastfeeding duration, days	349.50 (369.16)	0	
2. N = 99			
3. ≤30	1. 13%		
4. ≤90	2. 22%		
5. ≤180	3. 42%		
6. ≤365	4. 67%		
7. >365	5. 33%		
Girls/Boys	58/47	16/18	0.434
Years of age	8.76 (3.08)	9.27 (3.03)	0.405
Weight, kg	33.03 14.43	36.09 14.01	0.292
Weight percentile	53.99 (26.80)	55.24 (28.12)	0.818
Height, cm	133.64 (21.63)	140.00 (21.56)	0.153
Height percentile	62.78 (26.44)	63.10 (25.65)	0.953
Body mass index	17.22 (3.10)	18.59 (3.11)	0.069
Body mass index percentile	45.32 (26.01)	49.50 (29.79)	0.445
Heart rate, bpm	82.97 (14.07)	86.88 (16.08)	0.181
PR interval, msec	129.52 (18.54)	127.91 (16.37)	0.654
QRS duration, msec	83.59 (8.49)	84.67 (7.97)	0.487
QT interval, msec	363.32 (29.75)	359.30 (30.01)	0.501
QT_c_ interval, msec	423.38 (22.72)	427.85 (19.49)	0.310
P-wave axis, degrees	37.99 (24.53)	47.12 (17.09)	0.048 *
QRS-wave axis, degrees	58.65 (28.30)	55.15 (25.10)	0.526
T-wave axis, degrees	38.93 (18.80)	45.06 (11.56)	0.079

All values are expressed as mean (standard deviation) unless otherwise stated. Abbreviations: bpm, beats per minute; ECG, electrocardiographic; msec, millisecond; Qt_c_, corrected QT interval; * *p* < 0.05.

**Table 2 children-10-01255-t002:** Pair wise Pearson’s correlation coefficient R (*p*-value), between ECG parameters near significant link with feeding type, and the percentiles of the somatometric characteristics of children.

	Percentiles of Somatometric Characteristics		
ECG Parameters	Weight	Height	Body Mass Index
Heart rate	−0.036 (0.683)	−0.108 (0.219)	0.021 (0.811)
P-wave	−0.106 (0.231)	−0.057 (0.231)	−0.050 (0.573)
T-wave	0.014 (0.873)	0.158 (0.071)	−0.100 (0.255)

All values are expressed as mean (standard deviation) unless otherwise stated. Abbreviations: ECG, electrocardiographic.

## Data Availability

All data are available from the corresponding author upon reasonable request.

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
