# Peer review of "P-Wave Axis of Schoolchildren Who Were Once Breastfed"

_children, 2023, doi:10.3390/children10071255_

Round 1
Reviewer 1 Report
This observational study presents limitations in terms of small case numbers, leading me to reject the paper. The following reasons support my decision:
-
1. The statistical method employed is inadequate. With only 33 children in the never breastfed group, the use of a t-test is questionable due to the small sample size. Non-parametric tests would be more appropriate in this scenario.
-
2. To establish a robust causal relationship, it is crucial to demonstrate a dose-dependent effect. Is there a correlation between breast feeding duration and the p axis? This information would strengthen the study's findings.
-
3. The authors' conclusion is primarily based on a borderline significant p-value, without accounting for potential confounding factors such as height, weight, and BMI. It is important to adjust for these confounders to ensure a more accurate interpretation of the results.
Author Response
Response to Reviewer 1
We appreciate the efforts of reviewer 1 in reading our manuscript and pointing out debatable points in it. Below we answer one after the other the questions raised by her/him. In the text we have marked additions in yellow and have crossed out with a line what we have deleted.
1. The statistical method employed is inadequate. With only 33 children in the never breastfed group, the use of a t-test is questionable due to the small sample size. Non-parametric tests would be more appropriate in this scenario.
ANSWER: The value of the Kolmogorov-Smirnow test statistic of the P-wave axis of our never breastfed children (D) is .15261.The p-value is .38695. Our data do not differ significantly from that which is normally distributed.
Regarding the P-wave axis of ever breastfed children, again our data do not differ significantly from that which is normally distributed. The value of the K-S test statistic (D) is .11097. The p-value is .13966.
By the way, we would like to draw attention to onee of the references that support our choice: “A Monte Carlo investigation evaluates the comparative power of the independent samples t-test and its nonparametric counterpart, the Wilcoxon Rank-Sum (WRS) test, to violations from population normality, using three commonly occurring distributions and small sample sizes. The t-test was more powerful under relatively symmetric distributions, although the magnitude of the differences was moderate. Under distributions with extreme skews, the WRS held large power advantages. When distributions consist of heavier tails or extreme skews, the WRS should be the test of choice.”
Bridge, P. D., & Sawilowsky, S. S. (1999). Increasing physicians' awareness of the impact of statistics on research outcomes: comparative power of the t-test and and Wilcoxon Rank-Sum test in small samples applied research. Journal of clinical epidemiology, 52(3), 229–235. https://doi.org/10.1016/s0895-4356(98)00168-1
Given the characteristics of the sample and the findings in the literature, we made the decision, which we continue to defend, that the t-test was quite appropriate for our study.
2. To establish a robust causal relationship, it is crucial to demonstrate a dose-dependent effect. Is there a correlation between breast feeding duration and the p axis? This information would strengthen the study's findings.
ANSWER: We added these Paragraphs to Results and Limitations, respectively:
On the other hand, we found no correlation between days of any breastfeeding and degrees of the P-wave axis, Pearson's correlation coefficient R = 0.061, p-value = 0.552.
Moreover, we found no correlation between days of any type of breastfeeding and degrees of the P-wave axis in schoolchildren when we know that a dose-response association would have strengthened a causal hypothesis.
3. The authors' conclusion is primarily based on a borderline significant p-value, without accounting for potential confounding factors such as height, weight, and BMI. It is important to adjust for these confounders to ensure a more accurate interpretation of the results.
We all know that P-values depend on the strength of the association and on the sample size. Frequently, the p-value is considered significant even though the effect is tiny if the sample size is very big. On the other hand, if the sample size is small as in our case, an effect may be significant yet still fall short of the p<0.05 threshold.
Reviewer 2 Report
General comment:
The authors investigated an observational cross-sectional study to evaluate the correlation between breastfeeding and ECG parameters in school-age children. They found that non-breastfed children tend to have a vertical atrial axis, and discussed the hypothesis to explain the mechanism of their result. The manuscript is adequately written, and the reviewer has only a few comments.
Specific comment 1:
Can the authors discuss why breastfeeding affects the atrial axis but not the ventricular axis in the manuscript?
Specific comment 2:
I think the conclusion part of the manuscript is not a conclusion obtained from the current study but only the discussion. Therefore, these contents should move to the discussion part to be exact.
Author Response
Response to Reviewer 2
RESPONSE: We appreciate how hard the reviewer has worked to help us improve this study. Below we respond to each of the questions that she/he asked us. In the text we have marked additions in yellow and have crossed out with a line what we have deleted.
General comment:
The authors investigated an observational cross-sectional study to evaluate the correlation between breastfeeding and ECG parameters in school-age children. They found that non-breastfed children tend to have a vertical atrial axis, and discussed the hypothesis to explain the mechanism of their result. The manuscript is adequately written, and the reviewer has only a few comments.
Specific comment 1:
Can the authors discuss why breastfeeding affects the atrial axis but not the ventricular axis in the manuscript?
RESPONSE: A possible explanation for our finding that breastfeeding is associated years later with horizontalization of the P-wave axis is that breastfed infants have less respiratory morbidity than never breastfed infants. For about 50 years, studies both on adults and children have documented that verticalization of the P-wave axis is consistently associated with obstructive respiratory pathology and that it has a linear relationship with altered FEV1. Although the exact aetiology of the vertical P-wave axis in these cases is unknown, it most likely stems from the hyperinflation of the lungs, which causes the heart to rotate into a more vertical posture. Research regarding the permanent effect of respiratory pathology on the ECG parameters has not found any abnormal clockwise rotation of the QRS axis; conversely, the P-wave axis may be a sensitive indicator of the rotation of the heart.
Siegler D. Reversible electrocardiographic changes in severe acute asthma. Thorax. 1977;32(3):328-32. doi: 10.1136/thx.32.3.328.
Yücel O, Yildiz M, Altinkaynak S, Sayan A. P-wave dispersion and P-wave duration in children with stable asthma bronchiale. Anadolu Kardiyol Derg. 2009;9(2):118-22.
Gordina AV, Egoshina KA, Eliseeva TI, Vinogradova NG, Ovsyannikov DY, Tush EV, Prakhov AV, Daniel-Abu MI, Khaletskaya OV, Kubysheva NI. The Relationship Between Bronchial Patency and Parameters of ECG Supraventricular Component in Children With Bronchial Asthma. Front Pediatr. 2020;8:576. doi: 10.3389/fped.2020.00576.
Specific comment 2:
I think the conclusion part of the manuscript is not a conclusion obtained from the current study but only the discussion. Therefore, these contents should move to the discussion part to be exact.
RESPONSE: We thank this comment. Conclusion now reads: “We report a significant clockwise rotation of above 15 grades in the mean P-wave axis of never breastfed schoolchildren compared to their ever-breastfed peers.” The rest of previous Conclusion has moved to Discussion.
Reviewer 3 Report
Please clarify if the "more vertical P-wave axis" concerning never breasrfed children, has clinical significancy or is pathological. Besides, we cannot clarify the mode of breastfeeding or the duration of it. Propably the results are not strong enough in case of minimum breastfeeding in the majority of children. In general, the sample propably shows heterogeneity without randomization and study a factor that is not clear that is pathological innon breastfeedind sample compared to breastfeeding children.
No serious problems.
Author Response
Response to Reviewer 3.
RESPONSE: We thank Reviewer 3 for taking the time to thoroughly study our manuscript and there is no doubt that his/her comments will lead to a clearer presentation of our study. Below we answer one after the other the questions raised by her/him. In the text we have marked additions in yellow and have crossed out with a line what we have deleted.
Comments and Suggestions for Authors:
Please clarify if the "more vertical P-wave axis" concerning never breastfed children, has clinical significancy or is pathological.
RESPONSE: We only report early stages of the P-wave axis clockwise rotation (within the norm) of never breastfed children, clearly ruling out serious events. These children do not (yet) exhibit the cardiovascular problems that may be seen in never breastfed adults.
The last paragraph of the corrected text now reads as follows: “In the last few years, research on pre-pathological findings is getting more significant in terms of Public Health. This includes slight, within normal ranges, but significant changes [28]”
Stuebe AM, Michels KB, Willett WC, Manson JE, Rexrode K, Rich-Edwards JW. Duration of lactation and incidence of myocardial infarction in middle to late adulthood. Am J Obstet Gynecol. 2009;200(2):138.e1-8. doi: 10.1016/j.ajog.2008.10.001.
Rich-Edwards JW, Stampfer MJ, Manson JE, Rosner B, Hu FB, Michels KB, Willett WC. Breastfeeding during infancy and the risk of cardiovascular disease in adulthood. Epidemiology. 2004;15(5):550-6. doi: 10.1097/01.ede.0000129513.69321.ba.
Besides, we cannot clarify the mode of breastfeeding or the duration of it. Probably the results are not strong enough in case of minimum breastfeeding in the majority of children.
RESPONSE: We have recognized that the lack of detailed information on the duration of the different types of infant feeding practices in this sample is one of our limitations. However, systematic reviews increasingly show the benefits of ever versus never breastfeeding, which could be explained by a biological threshold mechanism rather than a linear relationship.
Rich-Edwards JW, Stampfer MJ, Manson JE, Rosner B, Hu FB, Michels KB, Willett WC. Breastfeeding during infancy and the risk of cardiovascular disease in adulthood. Epidemiology. 2004;15(5):550-6. doi: 10.1097/01.ede.0000129513.69321.ba.
Güngör D, Nadaud P, Dreibelbis C, LaPergola C, Terry N, Wong YP, Abrams SA, Beker L, Jacobovits T, Järvinen KM, Nommsen-Rivers LA, O’Brien KO, Oken E, Pérez-Escamilla R, Ziegler E, Casavale KO, Spahn JM, Stoody E. Never Versus Ever Feeding Human Milk and Food Allergies, Allergic Rhinitis, Atopic Dermatitis, and Asthma: A Systematic Review [Internet]. Alexandria (VA): USDA Nutrition Evidence Systematic Review; 2019.
Güngör D, Nadaud P, Dreibelbis C, LaPergola C, Terry N, Wong YP, Abrams SA, Beker L, Jacobovits T, Järvinen KM, Nommsen-Rivers LA, O’Brien KO, Oken E, Pérez-Escamilla R, Ziegler E, Casavale KO, Spahn JM, Stoody E. Never Versus Ever Feeding Human Milk and Celiac Disease: A Systematic Review [Internet]. Alexandria (VA): USDA Nutrition Evidence Systematic Review; 2019.
In general, the sample probably shows heterogeneity without randomization and study a factor that is not clear that is pathological in non-breastfeeding sample compared to breastfeeding children.
RESPONSE. In this preliminary study we have no data to establish that our sample is heterogeneous or not. We report a rotation in the P-wave axis that is not pathological and that is associated with the type of infant feeding. In no way do we point to a cause-effect relationship between these two variables because this is a descriptive study, and furthermore the classical conditions for establishing such a relationship are not met.
Round 2
Reviewer 1 Report
Non-prarametric statistics revealed non-significnat results. As to which way is better for p axis statitstics is the field for statisticians. I don't think the results can support the authors' main conclusion.
Reviewer 3 Report
Thank you for your reply.